# Immune responses to a single dose of the AZD1222/Covishield vaccine in health care workers

Chandima Jeewandara[1,8], Achala Kamaladasa[1,8], Pradeep Darshana Pushpakumara[1,8], Deshni Jayathilaka[1,8], Inoka Sepali Aberathna[1], Danasekara Rallage Saubhagya Rasikangani Danasekara[1], Dinuka Guruge[2], Thushali Ranasinghe[1], Shashika Dayarathna[1], Thilagaraj Pathmanathan[1], Gayasha Somathilake[1], Panambara Arachchige Deshan Madhusanka[1], Shyrar Tanussiya Ramu[1], Tibutius Thanesh Pramanayagam Jayadas[1], Heshan Kuruppu[1], Ayesha Wijesinghe[1], Herath Mudiyanselage Thashmi Nimasha[1], Dushantha Milroy[3], Achini Anuja Nandasena[4], Poththawela Kankanam Gamage Nilanka Sanjeewani[4], Ruwan Wijayamuni[2], Sudath Samaraweera[5], Lisa Schimanski[6,7], T. K. Tan [6,7], Tao Dong [6,7], Graham S. Ogg[6], Alain Townsend [6,7] & Gathsaurie Neelika Malavige [1,6 ✉]

Several COVID-19 vaccines have received emergency approval. Here we assess the immunogenicity of a single dose of the AZD1222 vaccine, at one month, in a cohort of health care workers (HCWs) (629 naïve and 26 previously infected). 93.4% of naïve HCWs seroconverted, irrespective of age and gender. Haemagglutination test for antibodies to the receptor binding domain (RBD), surrogate neutralization assay (sVNT) and ex vivo IFNγ ELISpot assays were carried out in a sub-cohort. ACE2 blocking antibodies (measured by sVNT) were detected in 67/69 (97.1%) of naïve HCWs. Antibody levels to the RBD of the wild-type virus were higher than to RBD of B.1.1.7, and titres to B.1.351 were very low. Ex vivo T cell responses were observed in 30.8% to 61.7% in naïve HCWs. Previously infected HCWs, developed significantly higher (p < 0.0001) ACE2 blocking antibodies and antibodies to the RBD for the variants B.1.1.7 and B.1.351. This study shows high seroconversion after one vaccine dose, but also suggests that one vaccine dose may be insufficient to protect against emerging variants.

[1] Allergy Immunology and Cell Biology Unit, Department of Immunology and Molecular Medicine, University of Sri Jayewardenepura, Nugegoda, Sri Lanka. [2] Colombo Municipal Council, Colombo, Sri Lanka. [3] Base Hospital, Dambadeniya, Sri Lanka. [4] Department of Family Medicine, University of Sri Jayewardenepura, Sri Jayewardenepura Kotte, Sri Lanka. [5] Epidemiology Unit, Ministry of Health Sri Lanka, Colombo, Sri Lanka. [6] MRC Human Immunology Unit, MRC Weatherall Institute of Molecular Medicine, University of Oxford, Oxford, UK. [7] Centre for Translational Immunology, Chinese Academy of Medical Sciences Oxford Institute, University of Oxford, Oxford, UK. [8] These authors contributed equally: Chandima Jeewandara, Achala Kamaladasa, Pradeep Darshana Pushpakumara, Deshni Jayathilaka. ✉email: gathsaurie.malavige@ndm.ox.ac.uk

The first cases of COVID-19 due to infection with the SARS-CoV-2 infection were reported in December 2019, from Wuhan in the Hubei province in China[1]. However, within 1 year, not only were several types of vaccines for COVID-19 developed, but they were used in mass immunization campaigns in many parts of the world, after successful completion of phase 3 trials[2–4]. The mRNA COVID-19 vaccines Pfizer BioNTech received emergency use authorization on 11 December 2020 and the Moderna on the 18th of December in USA, while the UK MHRA approved the AstraZeneca vaccine on the 30th of December 2020 (refs. [2,4]). The mass scale immunization campaigns that were initiated in December and early January 2021 have already shown to be effective by significantly reducing deaths, severe disease and hospitalizations in groups that received these vaccines[5,6].

While most of the vaccines for prevention of COVID-19 are two-dose vaccines, some vaccines such as the Johnson and Johnson adenoviral vector vaccine comprise a single dose, reporting an efficacy rate of 66% against symptomatic infection and 85% efficacy against severe disease[7]. Although the efficacy of a single-dose administration of the other WHO-approved vaccines has not been evaluated in large clinical trials, in some countries, in order to administer the first dose to a larger population, the second dose was delayed for up to 12 weeks[8]. A single dose of both the BNT162b2 (Pfizer BioNTech) vaccine and the AZD1222 (Astrazeneca) adenoviral vector vaccine was found to significantly reduce hospitalizations due to COVID-19, 28–34 days since administration of the first dose[9]. It was recently shown that a single dose of the BNT162b2 (Pfizer BioNTech) vaccine induced T cell and antibody responses that were comparable to those who were naturally infected with the SARS-CoV-2, several weeks or months following infection[10]. Although these data suggest that in a pandemic situation, where most countries have a shortage of vaccines, administering a single dose of a two-dose vaccine, does indeed offer substantial protection, there has been criticism that such an approach would give rise to the emergence of variants, due to a suboptimum immune response in those who only receive a single dose of a vaccine[8,11]. Those especially with haematological malignancies were shown to have a suboptimal immune response to a single dose of the BNT162b2 (Pfizer BioNTech), which leave them vulnerable to infection with the SARS-CoV-2 and for potential emergence of new variants[12]. However, some countries such as Canada have decided to delay the second dose for 16 weeks, despite these concerns[13].

There have been many variants of concern which are due to mutations in the spike protein of the virus, which either increase disease transmission, evade detection by currently available diagnostics or the mutations are in major sites where neutralizing antibodies bind to, and, therefore, they have a potential to affect vaccine efficacy[14]. The B.1.1.7 variant, which was initially detected in the UK, has shown to associate with higher transmissibility and higher mortality rates[14,15]. Although AZD1222 and BNT162b2 (Pfizer BioNTech) have shown a slightly reduced neutralization activity against B.1.1.7, it did not have a significant impact on vaccine efficacy[16,17]. However, the E484K mutation present in both the B.1.351 variant and P.1 variant have shown to significantly affect the neutralizing ability of the antibodies generated by most vaccines[16–18]. Since most of the COVID-19 vaccines underwent clinical trials, when these particular variants were not dominant, it would be important to determine the immune responses generated by these vaccines in neutralizing these variants of concern.

Although many developed countries such as the UK, Europe and USA have administered one dose of a COVID-19 vaccine to over 15% of their population by 1st of April 2021, many South Asian and South East Asian countries have administered one dose for <5%, while some African and Asian countries have immunized <1%[19]. Therefore, many countries in the world would have a partially immunized population, with a single vaccine dose administered. Furthermore, due to recent concerns regarding possible side effects such as cerebral venous thrombosis and thrombocytopenia, in relation to the AZD1222 vaccine[20], many individuals in some countries appear to be hesitant to obtain the second dose. In order to determine the immunogenicity of a single dose of the AZD1222/Covishield vaccine in a real-time situation, we assessed the immunogenicity (antibody and T cell responses), in a large cohort of healthcare workers (HCWs) in Sri Lanka, who received the AZD1222/Covishield vaccine during late January/early February and we also assessed the immune responses generated by these vaccines against the variants of concern (B.1.1.7 and B.1.351).

## Results

The demographic characteristics and previous infection status of the 655 HCWs is shown in Table 1. In total, 26/655 (3.9%) of individuals had past infection with the SARS-CoV-2. The median age of the HCWs was 41 years (range 21–81 years). In all, 367 (57.9%) were females and 50 (7.9%) had at least one comorbid condition (hypertension, diabetes or chronic kidney disease). The overall seroconversion after a single dose was 588 (93.4%). The seropositivity of these individuals between 28 and 32 days since obtaining the first dose of the vaccine is shown in Table 1. The seroconversion rates were highest in the 40–49 age group, whereas the seroconversion rates were lower in those >60 years of age 81.6%. Seroconversion rates were equal among males (244, 92.8%) and females (343, 93.4%). There was no difference in the SARS-CoV-2 total antibody levels between males (median 7.1, IQR 3.57–11.47 antibody index) compared to females (median 7.7, IQR 4.1–11.81). Of the 50 individuals who had comorbidities

**Table 1 Seropositivity rates of a single dose of the ChadOx1 between 28 and 32 days in HCWs who were seronegative at baseline.**

| Age group | Seropositive | Seronegative | Antibody index (total antibodies) Median (IQR) |
|---|---|---|---|
| 20–29 (n = 100, males = 43) | 95 (95%) | 5 (5%) | 7.5 (4.3–10.7) |
| 30–39 (n = 182, males = 87) | 164 (90.1%) | 18 (9.9%) | 6.1 (3.3–10.9) |
| 40–49 (n = 160, males = 62) | 159 (98.7%) | 5 (3.1%) | 8.8 (4.6–12.1) |
| 50–59 (n = 149, males = 58) | 139 (93.2%) | 8 (5.4%) | 7.4 (3.3–14.7) |
| >60 n = 38, males = 11 | 31 (81.6%) | 7 (18.4%) | 8.1 (2.3–12.1) |

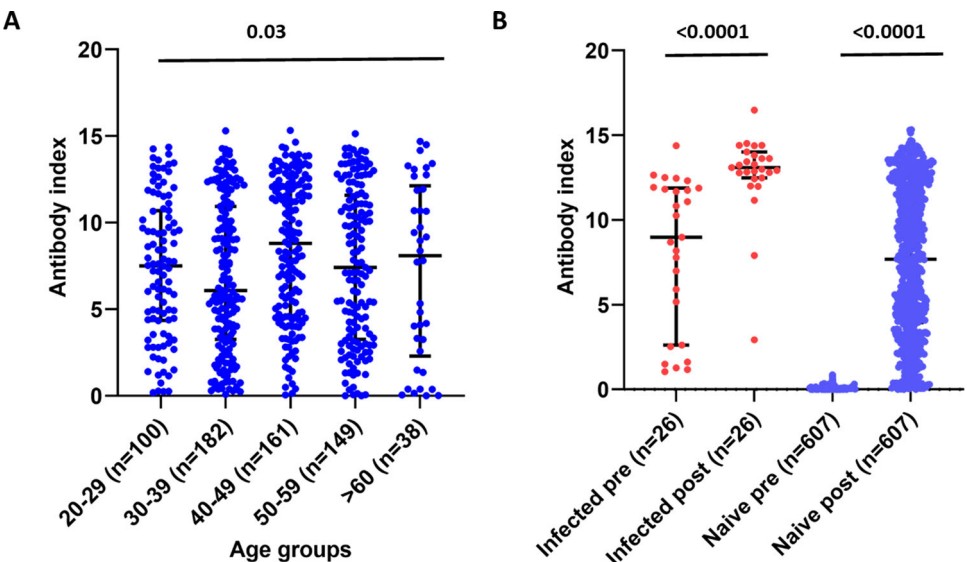

**Fig. 1 SARS-CoV-2-specific total antibody levels in vaccinated individuals.** SARS-CoV-2 total antibody levels (antibody index) in those in different age groups ($n = 629$) (**A**), and the total antibody levels in those who had previous infection at baseline and 28–32 days after a single dose ($n = 26$), and in SARS-CoV-2 uninfected individuals at baseline and 28–32 days after a single dose ($n = 629$) (**B**) were measured using the WANTAI SARS-CoV-2 Ab ELISA assay. The Kruskal–Wallis test was used to compare the differences in the antibody titres (antibody indexes) in each age group and the Wilcoxon matched-pairs signed-rank test was used to compare the means of the antibody indexes, before and after the vaccine. All tests were two sided. Data are presented as median values ± interquartile ranges as appropriate.

(those who had either diabetes, hypertension or chronic kidney disease), 48 (96%) seroconverted.

**Antibody titres in naïve individuals and those who were immune to the SARS-CoV-2.** The antibody index is an indirect measurement of the antibody levels of this SARS-CoV-2 total antibody assay. The median antibody titres were lowest in the 30- to 39-year-old age group (median 6.1, IQR = 3.3–10.9 index value), but the levels in >60 age group showed a median of 8.1 (IQR = 2.3–12.13 index value), which was comparable to other age groups. The overall antibody titres in different age groups was statistically significant ($p = 0.03$) when compared using the Kruskal–Wallis test (Fig. 1A).

Twenty-six individuals (females = 17) had a past COVID-19 infection, and only 6/26 had a symptomatic infection, between May 2020 and December 2020. Past SARS-CoV-2 infection with the other 20 individuals was determined following the detection of SARS-CoV-2-specific antibodies, with the Wantai total antibody assay, which has shown to be 100% specific in the Sri Lankan population. Therefore, they are likely to have had an asymptomatic infection. The antibody index values of those who had past COVID-19 at the time of recruitment was a median of 8.9 (IQR 2.6–11.9), which significantly ($p < 0.0001$) rose to a median of 13.1 (IQR12.5–14.0) between 28 and 32 days following a single dose of the vaccine (Fig. 1B).

**Antibodies to the receptor-binding domain of the spike protein, measured by the haemagglutination test.** Haemagglutination test (HAT) measures antibodies to the receptor-binding domain (RBD), where the RBD of the virus is linked to a nanobody IH4, specific for a conserved epitope within glycophorin A on red blood cells (RBCs)[21]. We have confirmed that this assay is negative in >99% of individuals prior to infection with SARS-CoV-2. We then used the assay to measure antibody titres to the RBD of the SARS-CoV-2 wild-type (WT) virus, B.1.1.7 variant and the B.1.351 variant in 68 individuals who were SARS-CoV-2 seronegative, and 26 individuals who had been infected with the

virus. The median post-vaccination HAT titres of those who were seronegative at baseline was 1:40 to the WT, 1:20 to B.1.1.7 and 0 to B.1.351, 28–32 days following a single dose of the vaccine (Fig. 2A). Following a single dose of the vaccine, those who had past COVID-19 had significantly higher HAT titres to the WT ($p < 0.0001$), B.1.1.7 ($p < 0.0001$) and the B.1.351 ($p < 0.0001$) (Fig. 2A). While the SARS-CoV-2 naïve individuals had significantly less ($p < 0.0001$) HAT titres to the B.1.1.7 compared to the WT following immunization, there was no significant differences in the HAT titres to WT and B.1.1.7 ($p = 0.21$) in those who were seropositive for SARS-CoV-2, at the baseline (Fig. 2A). Both groups of individuals who were seronegative or seropositive at baseline had significantly less ($p < 0.0001$) HAT titres to the B.1.351 compared to the WT and B.1.1.7 (Fig. 2A). Following a single dose of the vaccine, those who were seropositive at baseline, had a significant increase in the HAT titres for WT ($p = 0.005$), B.1.1.7 ($p < 0.0001$) and B.1.351 ($p = 0.0004$) (Fig. 2B).

A HAT titre of 1:20 was considered as positive for the presence RBD-specific antibodies. In all, 54/68 (78.2%) of individuals who were seronegative had positive RBD antibodies following a single dose of the vaccine. Forty-five (65.2%) had positive responses to the RBD of B.1.1.7 and 11 (15.9%) had responses to the RBD of B.1.351. At the baseline 21/26 (80.76%) who were known to be infected previously with the SARS-CoV-2 had antibodies to the RBD of the WT virus. In total, 19/26 (73%) had antibodies to RBD of B.1.1.7 and only 3/26 (11.5%) had antibodies to RBD of B.1.351. However, following a single dose of the vaccine, 25/26 (96.1%) developed antibodies to RBD of the WT, 25/26 (96.1%) to the RBD of B.1.1.7 and 20/26 (76.9%) to the RBD of B.1.351. There was no significant difference between HAT titres to the RBD of the WT, B.1.1.7 (Fig. 2C). However, there was a significance difference in the titres for the B.1.351 ($p = 0.006$), as those >60 years of age, had higher titres than some age groups (40–49 age group). This is possibly due to lower sample size in certain age groups. For instance, in the 40–49 age group ($n = 9$), no one had any antibodies to the RBD of B.1.351, whereas in the >60 age group ($n = 9$), five had IgG antibodies.

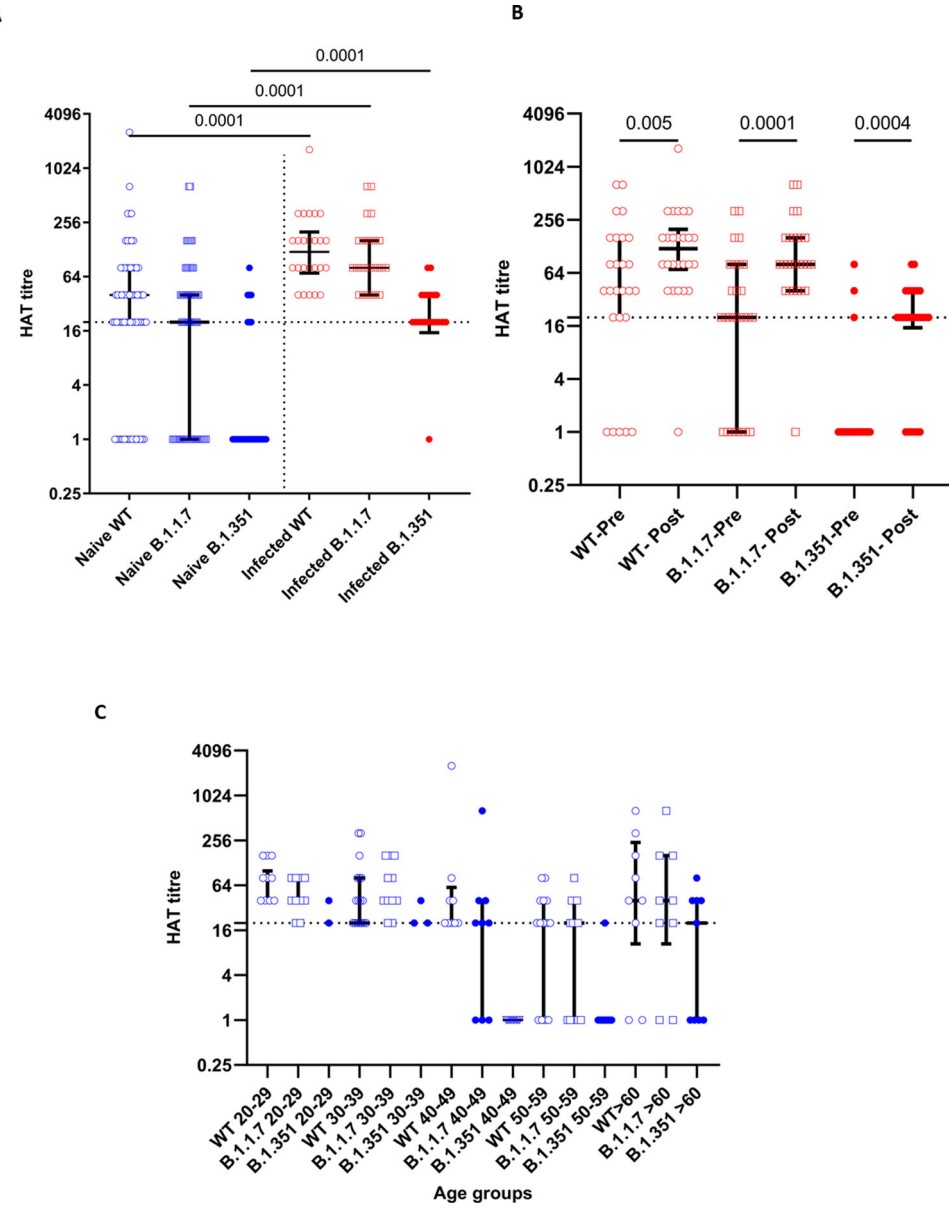

**Fig. 2 Haemagglutination test to detect antibodies to RBD of the wild type (WT), B.1.1.7 and B.1.351 in patients who were naïve and previously infected following a single dose of the AZD1222.** The HAT titres for the WT, B.1.1.7 and B.1.351 were measured in naïve individuals ($n = 68$) (blue) and previously infected individuals ($n = 26$) (red) 28–32 days following the vaccine (**A**). The HAT titres were measured in previously infected individuals at the baseline and following vaccination for the WT, B.1.1.7 and B.1.351 ($n = 26$) (**B**). The HAT titres were measured following a single dose in previously naïve individuals in different age groups (20–29 years = 13, 30–39 years = 25, 40–49 years = 8, 50–59 years = 13, >60 years = 9) (**C**). The black dotted line indicates the positive cut for the HAT. The Wilcoxon matched pairs signed-rank test was used to compare the means of the HAT titres before and after the vaccine. All tests were two sided. The Mann–Whitney $U$ test (two tailed) was used to calculate the differences in the means in the HAT titres in the infected and naïve individuals. Data are presented as median values ± interquartile ranges as appropriate.

**Surrogate neutralization assay to assess ACE2-blocking antibodies following a single dose of the AZD1222.** Due to the absence of facilities to carry out live virus assays to detect the presence of neutralizing antibodies (NAbs), we used a surrogate assay, which measured ACE2-blocking antibodies and has been shown to correlate with the NAbs specific for the SARS-CoV-2 (ref. [22]). The sVNT titres (percentage of inhibition of ACE2 binding) significantly increased 28–32 days post-vaccination in previously naïve individuals ($p < 0.0001$) and in previously infected individuals ($p < 0.0001$) (Fig. 3A). However, those who were previously infected with the SARS-CoV-2 (median 97.99, IQR 89.65–99.27% of inhibition) had significantly higher levels ($p < 0.0001$) than those who were naïve (median 69.42, IQR

54.09–81.54% of inhibition) following one dose of the vaccine (Fig. 3A). Only 2/68 (2.9%) individuals who were previously naïve failed to develop the level of 25% inhibition (regarded as "positive") following a single dose of the vaccine. Of those who were seropositive at recruitment, 6/26 (23.1%) were negative for the presence of ACE2-blocking antibodies by sVNT (<25% of inhibition). All such individuals developed a high level of ACE2-blocking antibodies following immunization.

The sVNT titres (ACE2-blocking antibodies) correlated significantly with the HAT titres for the WT virus (Spearman's $R = 0.71$, $p < 0.0001$), the B.1.1.7 (Spearman's $R = 0.67$, $p < 0.0001$) and with the B.1.351 (Spearman's $R = 0.50$, $p < 0.0001$) and with the SARS-CoV-2-specific total antibodies (Spearman's

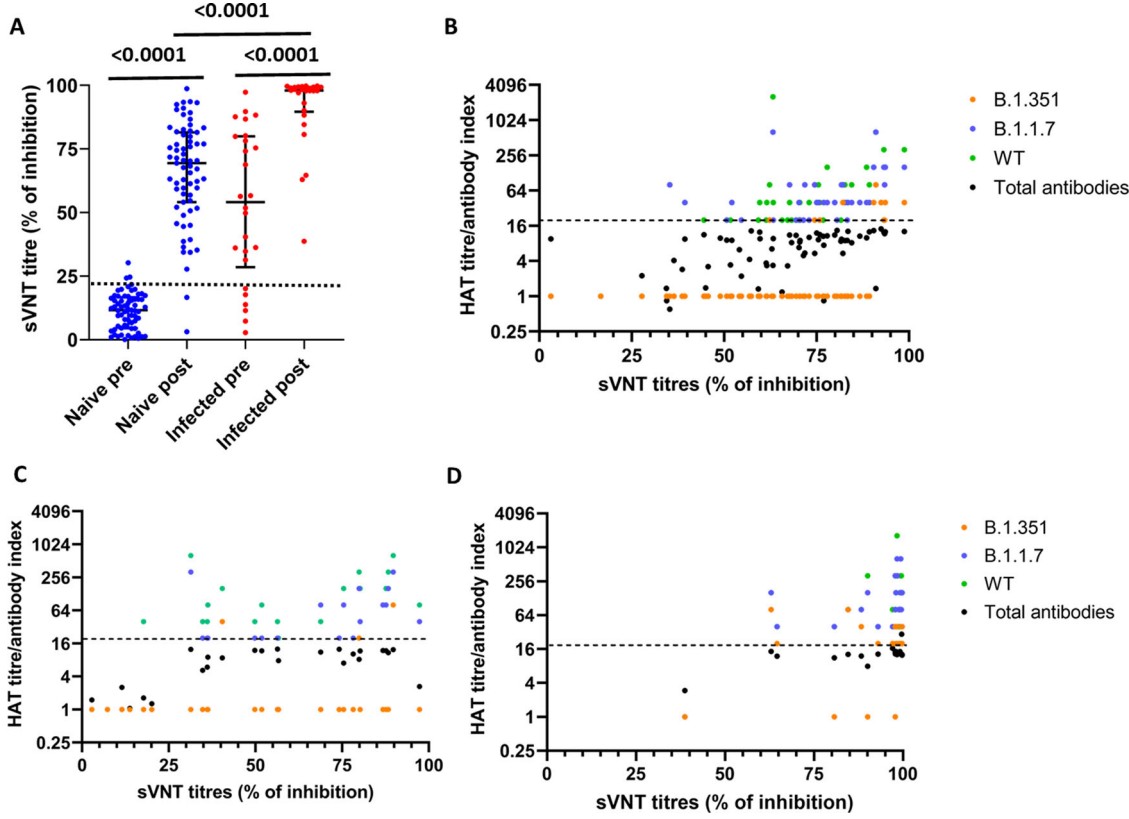

**Fig. 3 Surrogate SARS-CoV-2-neutralizing antibody assay (sVNT) in individuals who were naïve and previously infected following a single dose of the AZD1222 vaccine.** The sVNT titres (% of inhibition) were measured in naïve individuals ($n = 68$) (blue) and previously infected individuals ($n = 26$) (red) 28-32 days following the vaccine (**A**). The sVNT titres were correlated with the HAT titres for the WT virus (Spearman's $R = 0.71$, $p < 0.0001$), B.1.1.7 (Spearman's $R = 0.67$, $p < 0.0001$), B.1.351 (Spearman's $R = 0.51$, $p < 0.0001$) and the SARS-CoV-2-specific total antibodies (Spearman's $R = 0.54$, $p < 0.0001$) (**B**). The sVNT titres were correlated with the HAT titres for the WT virus (Spearman's $R = 0.64$, $p = 0.0005$), B.1.1.7 (Spearman's $R = 0.73$, $p < 0.0001$), B.1.351 (Spearman's $R = 0.25$, $p = 0.21$) and the SARS-CoV-2-specific total antibodies (Spearman's $R = 0.56$, $p = 0.003$) in previously infected individuals at baseline (**C**), and 28–32 days following a single dose of the vaccine for the WT virus (Spearman's $R = 0.47$, $p = 0.01$), B.1.1.7 (Spearman's $R = 0.36$, $p = 0.06$), B.1.351 (Spearman's $R = 0.13$, $p = 0.51$) and the SARS-CoV-2-specific total antibodies (Spearman's $R = 0.25$, $p = 0.21$) (**D**). The black dotted line indicates the positive cut-off for ACE2-blocking antibodies in **A** and for the HAT in **B–D**. The Wilcoxon matched-pairs signed-rank test was used to compare the means of the ACE-blocking antibodies (% of inhibition) before and after the vaccine. Spearman rank order correlation coefficient was used to evaluate the correlation between the HAT titres and the ACE-blocking antibodies. All tests were two sided. Data are presented as median values ± interquartile ranges as appropriate.

$R = 0.53$, $p < 0.0001$) in those who were previously naïve, suggesting that the ACE2-blocking antibodies and RBD antibodies increased similarly following the vaccine in these individuals (Fig. 3B). At the time of recruitment of those who were previously infected, the sVNT titres correlated significantly with the HAT titres for the WT virus (Spearman's $R = 0.63$, $p = 0.005$), and with the SARS-CoV-2-specific total antibodies (Spearman's $R = 0.56$, $p = 0.003$), and with B.1.1.7 (Spearman's $R = 0.73$, $p < 0.0001$) but not with B.1.351 (Spearman's $R = 0.25$, $p = 20$) (Fig. 3C). In these individuals, the sVNT titres significantly correlated with the HAT titres for the WT virus (Spearman's $R = 0.47$, $p = 0.01$), following vaccination but not with B.1.1.7 or B.1.351 or with the total antibodies post-vaccination (Fig. 3D).

Of the 68 naïve individuals, two individuals did not develop ACE2-blocking antibodies or antibodies to the RBD following vaccination, while 13 of those who did not appear to have detectable antibodies to the RBD by HAT, had ACE2-blocking antibodies. However, the ACE-blocking antibody titres were significantly less ($p < 0.0001$) in those who were negative by the HAT for antibodies (median 45, IQR 34.3–56.8% of inhibition), compared to those who were positive by the HAT assay (median 74.7, IQR 63.2–83.3% of inhibition).

In the previously infected individuals, the median HAT titres increased from a median of 40 (IQR 20–160) to a median of 120 (IQR 70–200) following a single dose of the vaccine. Interestingly, the increase was more for ACE2-blocking antibodies in previously infected individuals, which increased from 54.1 to 97.9%, suggesting that the increase of antibodies to the RBD is likely to be shifted towards the ACE2-blocking antibodies in those who were previously infected.

**Ex vivo T cell responses to overlapping peptides of the spike protein.** We investigated the ex vivo IFNγ ELISpot responses in 76 individuals to two overlapping pools representing the spike protein, S1 (peptide 1 to 130) and S2 (peptides 131 to 253). Of the 72 individuals, 4 individuals were previously infected with SARS-CoV-2. Of SARS-CoV-2 naïve individuals, only 2/68 had ex vivo T cell responses to the S1 pool of peptide pre-vaccination, possibly due to cross-reactivity with other seasonal coronaviruses. None of the naïve individuals had any responses to the S2 pool of peptides pre-vaccination. The ex vivo IFNγ ELISpot responses to both S1 and S2 significantly increased ($p < 0.0001$) (Fig. 4A). The responses to the S1 pool of peptides representing the early (peptide 1–130) region of the spike protein (median 397.5, IQR 165.0–702.5 spot-forming unit (SFU)/1 million peripheral blood

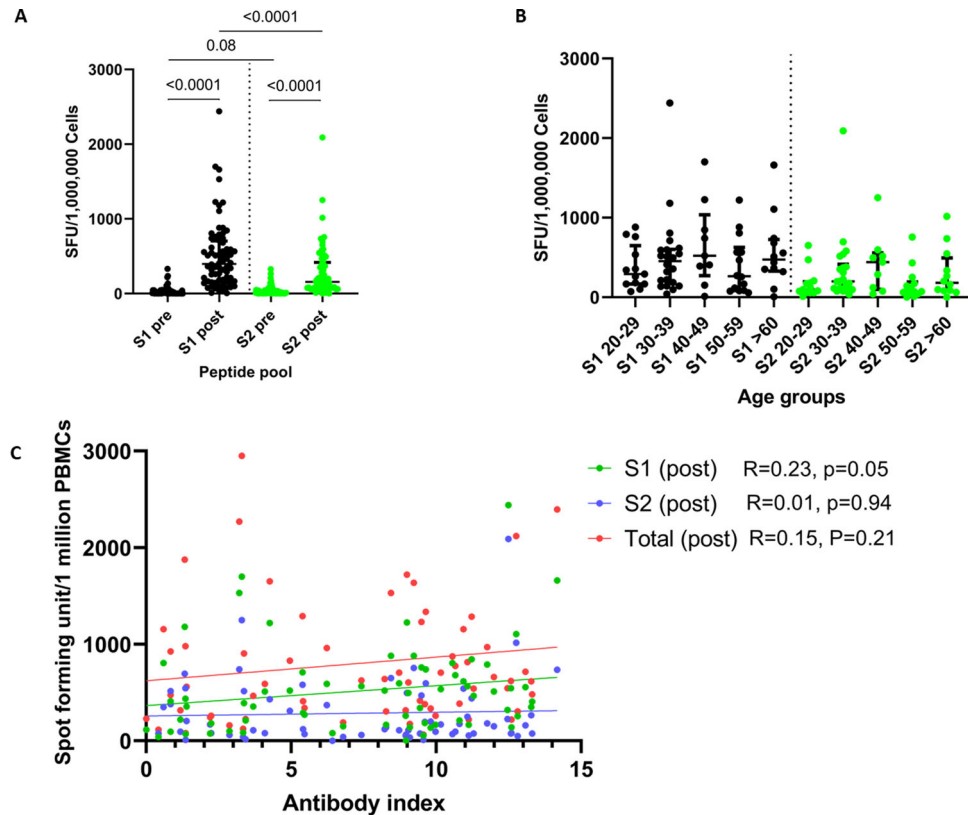

**Fig. 4 Ex vivo IFNγ ELISpot responses in individuals at baseline and 28–32 days following a single dose of the AZD1222/Covishield vaccine.** Ex vivo IFNγ ELISpot responses were measured to two pools representing the spike protein (S1 and S2) at the baseline (pre) and 28–32 days following the vaccine in total naïve individuals ($n = 72$) (**A**), and in different age sub-groups of naïve individuals ($n = 68$) (**B**). The association of the ex vivo IFNγ ELISpot responses to the two pools of the spike protein (S1 and S2) and the total responses to overlapping peptides of the spike protein did not correlate with the total antibody responses ($n = 68$) (**C**). The Wilcoxon matched-pairs signed-rank test was used to compare the means of ex vivo ELISpot responses to S1 and S2 pool of peptides before and after the vaccine. The Kruskal–Wallis test was used to compare the S1 and S2 ELISpot responses in different age groups. Spearman rank order correlation coefficient was used to evaluate the correlation between the ex vivo ELISpot responses and the SARS-CoV-2-specific antibodies (antibody index). All tests were two sided. Data are presented as median values ± interquartile ranges as appropriate.

mononuclear cells (PBMCs)) compared to the S2 pool (peptide 131–256) of overlapping peptides (median 155, IQR 75–417.5, SFU/1 million PBMCs). There were no significant differences to either S1 ($p = 0.57$) or S2 ($p = 0.06$), between the different age groups (Fig. 4B). A ex vivo ELISpot response of the mean±2 SD of the background responses was considered as a positive response. In all, 42/68 (61.7%) of individuals had responses to the S1 pool of peptides and 21/68 (30.8%) had a positive response to the S2 pool of peptides.

The ex vivo IFNγ ELISpot responses to S1, S2 or the total S protein did not correlate with the total antibody titres specific for SARS-CoV-2 (Fig. 4C). The ex vivo ELISpot responses also did not correlate with the HAT titres for the WT (Spearman's $R = -0.08$, $p = 0.48$) or with the % of inhibition (ACE-Abs) given by the sVNT assay (Spearman's $R = 0.02$, $p = 0.86$). There were no significant differences in the HAT titres in those who responded to S1 ($p = 0.34$) and S2 pool ($p = 0.86$) of peptides compared to those who did not respond to these peptides. There were also no significant differences in the ACE2-blocking antibodies (% of inhibition) in those who responded to S1 ($p = 0.66$) and S2 pools ($p = 0.42$) of peptides, compared to those who had no responses. One of the two individuals who had no antibody responses to the vaccine also did not generate any T cell responses, while the other person did have detectable T cell responses.

Of four individuals who were previously infected with SARS-CoV-2, three had a very low frequency of ex vivo IFNγ ELISpot responses pre-vaccination to both S1 (median 47.5, IQR

33.75–428.5 SFU/1 million PBMCs) and S2 (median 152.5, IQR 108.8–192.5 SFU/1 million PBMCs). The fourth individual was later found to have a recent infection (the individual had one day of mild fever, 14 days before to immunization). Following immunization, the frequency of ex vivo T cell responses increased several fold in those with past COVID-19.

## Discussion

In this study we have investigated antibody and ex vivo T cell responses to a single dose of the AZD1222 vaccine 28–32 days following immunization in previously naïve and infected HCWs. Our results show that 93.4% previously naïve individuals seroconverted to a single dose of the vaccine, irrespective of age and gender. A single dose of the vaccine was found to induce similar magnitude antibody and T cell responses in those who were <60 years of age and >60 years of age, although the seroconversion rates were lower in >60-year-olds compared to younger individuals. In naïve individuals, a single dose appeared to induce a higher proportion of ACE2-blocking antibodies than following natural infection. Our previous data in the Sri Lankan individuals with natural COVID-19 infection showed that although all individuals with moderate to severe illness had ACE2-Abs, assessed by the sVNT assay, 23/68 (33.3%) with mild illness did not have a response above the positive cut-off value (>25% of inhibition)[23]. In contrast, only 2/68 (2.9%) of previously naïve individuals failed to have a positive NAb following a single dose of the vaccine. Similar results were seen with the HAT assay

following natural infection and immunization. For instance, only 33/66 (50%) with those with asymptomatic/mild illness had a positive antibody response by the HAT assay for the WT virus at the end of 4 weeks (under review), whereas 78.2% had a positive response to the RBD antibodies by the HAT assay following a single dose of vaccine. Therefore, a single dose of the AZD1222 vaccine appears to induce a robust SARS-CoV-2 antibody response targeting the RBD of the virus, which is thought to associate with protection.

Individuals who have recently recovered from natural COVID-19 infection were shown to have robust CD4+ and CD8+ T cell responses to many of the viral proteins, which were of a higher magnitude and breadth in those who had experienced severe illness[24]. Eighteen to 32% of individuals were found to recognize different regions of the spike protein[24]. The T cell response frequencies were shown to be between 67 and 87% in individuals with mild illness in the convalescent phase or in exposed family members[25]. We found that 63.9% of individuals showed IFNγ ex vivo T cell responses to the S1 pool of overlapping peptides, following a single dose of the vaccine, which is comparable to what was seen following natural infection. The ex vivo ELISpot responses observed in our cohort following a single dose of the AZD1222 were slightly higher (median 397.5 for S1 pool and median 155 for S2 pool, SFU/1 million PBMCs) compared to the ex vivo ELISpot responses following a single dose of the BNT162b2 (Pfizer BioNTech) vaccine (median 58, SFU/1 million PBMCs)[10]. However, these are two different studies and, therefore, these variations could be due to assay variation between the laboratories, rather than a difference in the T cell responses induced by the two vaccines.

Although the World Health Organization and many other policy makers have recommended that those who have previously been infected with SARS-CoV-2 should obtain the vaccine[26], many such individuals have been hesitant. However, a single dose of the AZD1222 vaccine in previously exposed individuals not only significantly increased their ACE2-blocking antibodies but also significantly increased the RBD antibodies for the variants such the B.1.1.7 and B.1.351. The ACE2-blocking titres measured by the sVNT increased from a median of 54.1 to 97.9% of inhibition, in these individuals. Since a single dose resulted in a substantial increase in the ACE2-blocking titres and also the antibody responses to variants in previously infected individuals increased, it would be important to consider if a single dose of the vaccine would provide sufficient immunity in such individuals. In settings where the P1 or B.1.351 variants are causing severe disease even in individuals previously infected with the original SARS-CoV-2, our results suggest a single dose of vaccine based on the original sequence may still induce a significant increase in antibodies cross-reactive with the variants—perhaps sufficient to ameliorate disease. However, in naïve individuals, there was a significant reduction in the HAT titres to B.1.1.7, which could be due to reduction neutralization due to the single 501Y mutation in B.1.1.7. The significance of this in real-life situations could only be assessed by evaluating the vaccine efficacy for reducing asymptomatic and symptomatic in countries which have the WT (Wuhan virus strain) compared to the B.1.1.7 variant. However, given that B.1.1.7 appears to be the dominant variant in the majority of countries, this would be difficult to assess in the current scenario.

Two (2/68) naïve individuals did not have any responses to the vaccine (antibodies to RBD and ACE2-blocking antibodies), while one of these individuals had T cell responses. However, of the whole cohort of individuals 7.1% (43/607), had no detectable antibodies by the Wantai total antibody ELISA, which detects IgM, IgA and IgG to the RBD, while 21.8% were negative by HAT. Except for the seroconversion rates being lower in

individuals >60 years of age (7/43 who did not seroconvert), comorbidities did not affect seroconversion. It would be important to find out if these individuals who had a poor serological response to the vaccine would be more susceptible to infection in future in prospective studies.

In summary, a single dose of the AZD1222 vaccine induced high levels of antibodies to the RBD and ACE2-blocking antibodies, in previously naïve individuals, which was greater than immune responses in those who experience a mild or asymptomatic natural infection. The T cell responses were comparable to those following natural infection. In those who previously had COVID-19, a single dose induced very high levels of ACE2-blocking antibodies and antibodies to RBDs of SARS-CoV-2 variants of concern.

## Methods

Six hundred and fifty-five HCWs, who received their first dose of the AZD1222/Covisheild vaccine between the 29th January and 5th of February 2021, were included in the study following informed written consent. Demographic details such as age, gender, comorbid illnesses were recorded. The presence of comorbidities such as diabetes, hypertension, cardiovascular disease and chronic kidney disease was determined by a self-administered questionnaire at the time of recruitment. Blood samples were obtained from all individuals to determine the SARS-CoV-2 serostatus at baseline, while T cell study were carried out in only 72 individuals, who were randomly selected. A second blood sample was obtained between 28 and 32 days following the first dose to assess SARS-CoV-2-specific antibody and T cell responses. Ethics approval was obtained from the Ethics Review Committee of University of Sri Jayewardenepura. None of the individuals included in this study reported any COVID-19 infection during this one month.

**Detection of total antibodies to SARS-CoV-2.** SARS-COV-2-specific total antibody (IgM, IgG and IgA) responses were assessed using Wantai SARS-CoV-2 Ab ELISA (Beijing Wantai Biological Pharmacy Enterprise, China). A cut-off value for each ELISA was calculated according to the manufacturer's instructions. Based on the cutoff value, the antibody index was calculated by dividing the absorbance of each sample by the cutoff value, according to the manufacturer's instructions. This assay was shown to have a sensitivity of 98%[27] and was found to be 100% specific in serum samples obtained in 2018, in Sri Lankan individuals.

**HAT to detect antibodies to the RBD.** The HAT was carried out as previously described[21]. The B.1.1.7 (N501Y) and B.1.351 (N501Y, E484K, K417N) versions of the IH4-RBD reagent were produced as described[21], but included the relevant amino acid changes introduced by site-directed mutagenesis. These variants were titrated in a control HAT with the monoclonal antibody EY-6A (to a conserved class 4 epitope[21,28]) and found to titrate identically with the original version so 100 ng (50 μl of 2 μg/ml stock solution) was used for developing the HAT. Briefly, red blood cells from an O-negative donor were mixed with the IH4-RBD (a nanobody against a conserved glycophorin A epitope on red cells, linked to the RBD of SARS-CoV-2) and incubated for 1 h with serum. Phosphate-buffered saline was used as a negative control. At the end of the incubation the plate was tilted for 20 s and then photographed. The photograph of the plate was read by two independent readers to examine the "teardrop" formation indicative of a negative result. A complete absence of "teardrop" formation was scored as positive, and any flow of "teardrop" was scored as negative. The HAT titration was performed using 11 doubling dilutions of serum from 1:20 to 1:20480, to determine presence of RBD-specific antibodies. The RBD-specific antibody titres for the serum sample was defined by the last well in which the complete absence of "teardrop" formation was observed. RBD-specific antibody titres were also evaluated for the RBD of the B.1.1.7 variant and the B.1.351 variant in 69 individuals (69/72 seronegative individuals in whom T cell assays were carried out), who were seronegative at the baseline and in 26 who had previous infection with SARS-CoV-2.

**Measuring the presence of neutralizing antibodies to the SARS-CoV-2 using a surrogate assay.** Due to the lack of a BSL-3 facility to assess the presence of neutralizing antibodies, we adopted a recently developed surrogate virus neutralization test (sVNT)[22], which measures the percentage of inhibition of binding of the RBD of the S protein to recombinant ACE2 (Genscript Biotech, USA). Inhibition percentage ≥25% in a sample was considered as positive for ACE2-blocking antibodies. This assay was found to be 100% specific for measuring ACE2-blocking antibodies in the Sri Lankan population[23]. These assays were carried out on 69 individuals who were seronegative (69/72 seronegative individuals in whom T cell assays were carried out), and 26 individuals who were found to have past infection with SARS-CoV-2.

**Ex vivo ELISpot assay.** Ex vivo IFNγ ELISpot assays were carried out using freshly isolated PBMC obtained from 72 individuals at the time of recruitment and 28–32 days later[29]. Individuals for T cell assays were randomly recruited from the study participants, and we included those who consented to provide an additional blood volume for T cell assays (7 ml), in addition to the antibody assays (5 ml). Two pools of overlapping peptides named S1 (peptide 1–130) and S2 (peptide 131–253) covering the whole spike protein (253 overlapping peptides) were added at a final concentration of 10 μM and incubated overnight as previously described[24,30]. All peptide sequences were derived from the wild-type consensus and were tested in duplicate. PHA was included as a positive control of cytokine stimulation and media alone was applied to the PBMCs as a negative control. Briefly, ELISpot plates (Millipore Corp., Bedford, USA) were coated with anti-human IFNγ antibody overnight (Mabtech, Sweden). The plates were incubated overnight at 37 °C and 5% $CO_2$. The cells were removed, and the plates developed with a second biotinylated Ab to human IFNγ and washed a further six times. The plates were developed with streptavidin-alkaline phosphatase (Mabtech AB) and colorimetric substrate, The spots were enumerated using an automated ELISpot reader (AID Germany). Background (PBMCs plus media alone) was subtracted and data expressed as the number of SFU per $10^6$ PBMCs. A positive response was defined as mean ± 2SD of the background responses.

**Statistical analysis.** GraphPad Prism version 6 was used for statistical analysis. As the data were not normally distributed, differences in means were compared using the Mann–Whitney $U$ test (two tailed), and the Wilcoxon matched-pairs signed-rank test was used when comparing paired data. The Kruskal–Wallis test was used to compare the differences of the antibody levels and ex vivo ELISpot responses in different age groups. Spearman rank order correlation coefficient was used to evaluate the correlation between variables including the association between SARS-CoV-2-specific T cell responses, age and antibody responses.

**Statistics and reproducibility.** All ex vivo ELISpot assays were carried out in duplicate, with the relevant positive and negative controls to ensure, reproducibility. All the antibody assays (Wantai total antibody assay, ACE2 antibody-blocking assay/sVNT and HAT) was validated using blood samples collected in 2017 and 2018 ($n = 110$) to determine the specificity and the sensitivity were determined in serial blood samples taken from individuals with acute COVID-19 illness[23]. For the HAT, the photograph of the plate was read by two independent readers to examine the "teardrop" formation indicative of a negative result, in order to ensure reproducibility. During this study, all the relevant positive and negative controls were included in each assay.

**Study approvals.** Ethics approval was obtained from the Ethics Review Committee of University of Sri Jayewardenepura (COVID 01/21). The study was also approved by the Education, Training and Research unit of the Ministry of Health, Sri Lanka.

**Reporting summary.** Further information on research design is available in the Nature Research Reporting Summary linked to this article.

## Data availability

All data are available within the manuscript, figures and the tables. Individual data points are shown in all figures. Source data are provided with this paper.

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

## Acknowledgements

We are grateful to the World Health Organization, UK Medical Research Council and the Foreign and Commonwealth Office for support. T.K.T. is funded by the Townsend-Jeantet Charitable Trust (charity number 1011770) and the EPA Cephalosporin Early Career Researcher Fund. A.T. are funded by the Chinese Academy of Medical Sciences (CAMS) Innovation Fund for Medical Science (CIFMS), China (grant no. 2018-I2M-2-002).

## Author contributions

C.J., A.T. and G.N.M.: conceptualization. C.J., A.K., P.D.P., D.J. and L.S.: methodology. A.K., P.D.P., D.J. and G.N.M.: formal analysis. A.K., P.D.P. D.J., S.D., I.S.A., D. Madushanka, S.T.,

T.T.J., A.W., N.Y., T.P. and N.T.: investigation. I.S.A., S.D., T.R., G.S., H.K., A.N., D. Milroy and N.S.: data curation. C.J., D.G., R.W., S.S. and G.N.M.: project administration. C.J., G.N.M., G.S.O., A.T., T.D. and T.K.T.: funding acquisition. G.N.M. and G.S.O.: writing original draft. G.N.M., G.S.O., A.T., T.K.T. and T.D.: writing-review and editing. L.S. and T.K.T.: validation.

## Competing interests

G.N.M. is in the National Medicinal Regulatory Authority on the expert advisory panel in COVID-19 vaccines. S.S. in the Chief Epidemiologist in Sri Lanka and is involved in deciding vaccine priority lists.
