## [Peer Review File · Nature Communications]

Reviewers' Comments:

Reviewer #1:

Remarks to the Author:

NCOMMS-21-15309

In this manuscript, Jeewandara et al. evaluated the immunogenicity of the AZD1222 vaccine (by AstraZeneca) in a large cohort of health care workers in Sri Lanka. The cohort included individuals with different immune status (naïve or previously exposed to Covid-19). The authors assessed the seroconversion at early time point (30 days after vaccination), the antibody response (antibody titers to the RBD and neutralization) and the cellular immunity to the Spike protein after a single dose of AZD1222 vaccine. They showed that a single dose of vaccine was able to elicit SARS-COV2 antibody responses targeting the RBD of the virus, which could be associated with protection after 28 days independently of the immune status, moreover pre-exposed covid-19 individuals showed significantly higher immunogenicity compared to naïve individuals. Overall, the authors addressed an important question about the immunogenicity of a single dose vaccination regimen as many countries would have a partially immunized population with different immune status to Covid-19. It is crucial to provide data on immunity to one vaccine dose as well as to study vaccinees with prior SARS-COV2 who were not widely included in efficacy trials. As many variants surge it is important to know the immunogenicity of a vaccine against those variants.

The authors should clarify and add more information to improve the manuscript.

The cohort of health care workers used by the authors is not well characterized, some details are missing in table 1. The authors should add more information, for example about the % of female/male per age group, median of months from previous SARS-COV2 infection to vaccination. How was the immune status of pre-exposed individuals was determined? More details are needed about the disease in that group of individuals: % of symptomatic/ Asymptomatic/ hospitalizations/ mild illness/severe. How the pre-immunity is defined? Antibody titers? How is the comorbidity defined in this study (L300)? Please specify.

Due to the number of participants in the study, parametric tests could be more accurate and could give greater statistical power. Please check with statisticians if the statistical tests used in the study are accurate.

Figure 1A: For the statistic, what groups are compared in the figure? The authors should clarify the figure.

Line 139: The authors used 69 individuals who were SARS-COV2 negative to measure the HAT titers, how was this cohort of 69 individuals selected? Was the selection random? More details about those individuals should be added. The selection of these individuals could have an impact on the results. Similarly, the reader needs more details about the cohort of 76 individuals used for the T cell response. How was the selection done?

Figure 3A: Is there any significant difference in the sVNT titers between the group of naïve and infected individuals after vaccination?

In figure 3, the authors presented the sVNT titers (percentage of inhibition of ACE2 binding) against WT virus as a measurement of neutralization. To strengthen the manuscript, the authors should assess the "neutralizing" activity of vaccinees against the variants (B.1.1.7 and B.1.351 as done for the HAT titers. Any correlation between the HAT titers and the sVNT titers for the variants?

Figure 4C: For clarity, the values of the correlation coefficient and p should be added to the figure.

Figure legends: For all figures, the authors should add the number of participants in the legend mostly when it is not mentioned on the graph.

Line 272-278: The authors compared the immunogenicity of AZD1222 and BNT162b2 vaccines, although these data are from two different studies. The comparison is not accurate and the authors should revise the discussion about that point.

Typos:

L135: Delete "s"

L156-161: Mention "figure 2 B", this section is confusing without referencing the figures.

Reviewer #2:

Remarks to the Author:

The manuscript by Jeewandara and colleagues describe the immunity of the AstraZeneca Covidshield vaccine in cohort of health care workers. The authors use a combination of antibody and T cell assays to establish broad seroconversion and the elicitation of a degree of T cell immunity after a single dose. Some of the methods are written a little unclearly, as is the description of the results. The question of the utility of single dose immunisation from a public health standpoint is of interest to inform vaccine rollouts. However we already know a single dose of many of the vaccines is immunogenic from phase I studies and others (and even protective in some epidemiological assessments). Ultimately the results presented here are largely confirmatory and the manuscript might be better suited for a more specialised immunology journal.

I have some comments below:

1 - Table 1 appears incorrect or incomplete. There is no provision of relevant demographic data nor baseline seropositivity data for the cohort that were immunised.

2 - Figure 1 – unclear what an antibody index is and how this is calculated. This needs to be detailed in the Methods and the figure legends.

3 - Figure 2C – unclear why the number of values differ in the figure between RBD variants tested. Were not all samples run for each variant?

4 - Line239 - Some of the baseline seropositive subjects identified had acute infection? These and similar important details need to be included in the summary table of subjects.

5 - The major drop in antibody binding to B.1.1.7 using the HAT assay is somewhat unexpected, given only a single 501Y mutation in the RBD domain that differs from WT. What do the authors propose is causing this drop given plentiful other studies showing limited neutralisation escape for B.1.1.7. Is this just a loss of binding antibody?

Minor comments:

Some copyediting for readability would be appropriate.

Line 513 -Figure 1 - Kruskal-Wallis tests does not assess means.

Reviewer 1

In this manuscript, Jeewandara et al. evaluated the immunogenicity of the AZD1222 vaccine (by AstraZeneca) in a large cohort of health care workers in Sri Lanka. The cohort included individuals with different immune status (naïve or previously exposed to Covid-19). The authors assessed the seroconversion at early time point (30 days after vaccination), the antibody response (antibody titers to the RBD and neutralization) and the cellular immunity to the Spike protein after a single dose of AZD1222 vaccine. They showed that a single dose of vaccine was able to elicit SARS-COV2 antibody responses targeting the RBD of the virus, which could be associated with protection after 28 days independently of the immune status, moreover pre-exposed covid-19 individuals showed significantly higher immunogenicity compared to naïve individuals. Overall, the authors addressed an important question about the immunogenicity of a single dose vaccination regimen as many countries would have a partially immunized population with different immune status to Covid-19. It is crucial to provide data on immunity to one vaccine dose as well as to study vaccinees with prior SARS-COV2 who were not widely included in efficacy trials. As many variants surge it is important to know the immunogenicity of a vaccine against those variants. The authors should clarify and add more information to improve the manuscript.

Response: thank you for these positive comments.

Comment 1: The cohort of health care workers used by the authors is not well characterized, some details are missing in table 1. The authors should add more information, for example about the % of female/male per age group, median of months from previous SARS-COV2 infection to vaccination. How was the immune status of pre-exposed individuals was determined? More details are needed about the disease in that group of individuals: % of symptomatic/ Asymptomatic/ hospitalizations/ mild illness/severe. How is the pre-immunity defined? Antibody titers?

Response: We thank the reviewer for these very important questions. We have included the information of the gender in different age groups and the disease severity in the 26 individuals who were found to be infected with the SARS-CoV-2 at the time of immunization in Table 1. None of the 26 individuals had severe illness, but 6 had mild, symptomatic illness. The other 20 individuals were only found to be infected by the detection of SARS-CoV-2 specific antibodies by the antibody assays. This information is included in the revised Results section of the manuscript.

Comment 2: How is the comorbidity defined in this study (L300)? Please specify.

Response: The presence of diabetes, hypertension, cardiovascular disease and chronic kidney disease was determined by a self-administered questionnaire at the time of recruitment to the study. No tests were carried out to determine the presence of the above comorbidities. This has been clarified in the revised manuscript.

Comment 3: Due to the number of participants in the study, parametric tests could be more accurate and could give greater statistical power. Please check with statisticians if the statistical tests used in the study are accurate.

Response: Thank you for this question. We do have a large sample size for the seronegatives as baseline, but only a smaller sample size for those who were seropositive (infected) at baseline. In addition, for the T cell studies, HAT and the ACE2 blocking antibody assays, we were able to study 69 individuals and we found that data were not normally distributed. Although we could use parametric tests to carry out statistics on the large sample of seronegatives, it is more appropriate to carry out non-parametric tests throughout; and we have discussed these approaches with our local statistician.

Comment 4: Figure 1A: For the statistic, what groups are compared in the figure? The authors should clarify the figure.

Response: we apologize for lack of clarity. The differences in the antibody titres (antibody index values) of different age groups was compared with each other using the kruskal-wallis test. We have clarified this in the revised version of the manuscript.

Comment 5: Line 139: The authors used 69 individuals who were SARS-COV2 negative to measure the HAT titers, how was this cohort of 69 individuals selected? Was the selection random? More details about those individuals should be added. The selection of these individuals could have an impact on the results. Similarly, the reader needs more details about the cohort of 76 individuals used for the T cell response. How was the selection done?

Response: Again, we apologise for the lack of clarity. These individuals were recruited randomly based on those who gave consent to an additional volume of blood to be taken for T cell studies. Of these 76 individuals, 4 were found to have been previously infected with the SARS-CoV-2 virus. Of the remaining 72 individuals, only 69 individuals had adequate

amount of serum left for the sVNT assay (ACE2 blocking antibody assay) and the HAT assays, in which dilutions were carried out to determine the HAT titre for WT, B.1.1.7 and B.1.351. We have included this information in the revised version of the manuscript.

Comment 6: Figure 3A: Is there any significant difference in the titers between the group of naïve and infected individuals after vaccination?

Response: Those who were previously infected with the SARS-CoV-2 (median 97.99, IQR 89.65 to 99.27 % of inhibition) had significantly higher levels ($p < 0.0001$) than those who were naïve (median 69.42, IQR 54.09 to 81.54 % of inhibition). We have included these significance levels in the figure itself in addition to the text in the revised version of the manuscript.

Comment 7: In figure 3, the authors presented the sVNT titers (percentage of inhibition of ACE2 binding) against WT virus as a measurement of neutralization. To strengthen the manuscript, the authors should assess the “neutralizing” activity of vaccinees against the variants (B.1.1.7 and B.1.351 as done for the HAT titers. Any correlation between the HAT titers and the sVNT titers for the variants?

Response: Thank you for this question. We have included these data in the revised version of the manuscript. We have included the details of the results correlating the sVNT titres with the HAT B.1.1.7 and B.1.351 for naïve and previously infected individuals.

Comment 8: Figure 4C: For clarity, the values of the correlation coefficient and p should be added to the figure.

Response: Thank you for this suggestion. We have added this to the figure in the revised manuscript.

Comment 9: Figure legends: For all figures, the authors should add the number of participants in the legend mostly when it is not mentioned on the graph.

Response: Thank you for this suggestion. We have added this to the figure in the revised manuscript and have included the numbers in the legend itself.

Comment 10: Line 272-278: The authors compared the immunogenicity of AZD1222 and BNT162b2 vaccines, although these data are from two different studies. The

comparison is not accurate and the authors should revise the discussion about that point.

Response: Thank you for this very valid comment. We have changed the discussion mentioning that these are from two different studies, although the peptides that were used were from the same source and the same technique/protocol was used.

Comment 11: Typos:

L135: Delete “s”

L156-161: Mention “figure 2 B”, this section is confusing without referencing the figures.

Response: Thank you for these comments. We have corrected these typos.

Reviewer #2 (Remarks to the Author):

The manuscript by Jeewandara and colleagues describe the immunity of the AstraZeneca Covidshield vaccine in cohort of health care workers. The authors use a combination of antibody and T cell assays to establish broad seroconversion and the elicitation of a degree of T cell immunity after a single dose. Some of the methods are written a little unclearly, as is the description of the results. The question of the utility of single dose immunisation from a public health standpoint is of interest to inform vaccine rollouts. However we already know a single dose of many of the vaccines is immunogenic from phase I studies and others (and even protective in some epidemiological assessments). Ultimately the results presented here are largely confirmatory and the manuscript might be better suited for a more specialised immunology journal. I have some comments below:

Response: thank you for these positive comments.

Comment 1 - Table 1 appears incorrect or incomplete. There is no provision of relevant demographic data nor baseline seropositivity data for the cohort that were immunized.

Response: We thank the reviewer for this comment. We have included demographic data such as the number of males and females in each age group. In addition, table 1, includes the seroconversion rates of the health care workers who were seronegative at baseline. We have also added more information regarding those who were seropositive at baseline and regarding

comorbid illnesses in the study participants, in the revised version of Table 1 and the manuscript text.

Comment 2 - Figure 1 – unclear what an antibody index is and how this is calculated. This needs to be detailed in the Methods and the figure legends.

Response: We apologise for not including this information. We have clarified this in the methods section in the revised version of the manuscript.

Comment 3 - Figure 2C – unclear why the number of values differ in the figure between RBD variants tested. Were not all samples run for each variant?

Response: We thank the reviewer for this very valid question. We did run all the samples for the different variants. 2C is the responses to different variants in the different age groups. We have included the exact numbers that have been included in the figure legends in the revised version of the manuscript.

Comment 4 – Line239 - Some of the baseline seropositive subjects identified had acute infection? These and similar important details need to be included in the summary table of subjects.

Response: We apologize for lack of clarity. None of the baseline seropositives had an acute infection. The data regarding the baseline seropositive/ those who were previously infected are included in the revised version of the manuscript.

Comment 5 - The major drop in antibody binding to B.1.1.7 using the HAT assay is somewhat unexpected, given only a single 501Y mutation in the RBD domain that differs from WT. What do the authors propose is causing this drop given plentiful other studies showing limited neutralisation escape for B.1.1.7. Is this just a loss of binding antibody?

Response: Thank you for this very important question. The drop in the antibody binding to B.1.1.7 could be due to this limited neutralization escape for B.1.1.7. How this translates to real-life situations could only be assessed by evaluating the vaccine efficacy for reducing asymptomatic and symptomatic in countries which have the WT (Wuhan virus strain) compared to the B.1.1.7 variant. However, given that B.1.1.7 appears to be the dominant

variant in the majority of countries, this would be difficult to assess in the current scenario. We have discussed this in the revised version of the manuscript. We have added text to this point.

Comment 6: Minor comments: Some copyediting for readability would be appropriate.

Response: We apologise for any copy-editing improvements needed. We have carefully gone through the manuscript and adjusted text to help readability.

Line 513 -Figure 1 - Kruskal-Wallis tests does not assess means.

Response: Thank you for pointing out this mistake. We have corrected this in the revised version of the manuscript.

Thank you for considering our manuscript with Nature Communications.

Yours Sincerely,

Prof. Neelika Malavige

Reviewers' Comments:

Reviewer #1:

Remarks to the Author:

The authors addressed all of my comments and add more clarifications to the manuscript.

Reviewer #2:

Remarks to the Author:

The authors have addressed my comments and the manuscript has improved in terms of disclosure and clarity of relevant methods and results.